# Reaching the precision limit with tensor-based wavefront shaping

Rodrigo Gutiérrez-Cuevas [1] ✉, Dorian Bouchet [2], Julien de Rosny[1] & Sébastien M. Popoff [1]

Perturbations in complex media, due to their own dynamical evolution or to external effects, are often seen as detrimental. Therefore, a common strategy, especially for telecommunication and imaging applications, is to limit the sensitivity to those perturbations in order to avoid them. Here, instead, we consider enhancing the interaction between light and perturbations to produce the largest change in the output intensity distribution. Our work hinges on the use of tensor-based techniques, presently at the forefront of machine learning explorations, to study intensity-based measurements where its quadratic relationship to the field prevents the use of standard matrix methods. With this tensor-based framework, we can identify the maximum-information intensity channel which maximizes the change in its output intensity distribution and the Fisher information encoded in it about a given perturbation. We further demonstrate experimentally its superiority for robust and precise sensing applications. Additionally, we derive the appropriate strategy to reach the precision limit for intensity-based measurements, leading to an increase in Fisher information by more than four orders of magnitude compared to the mean for random wavefronts when measured with the pixels of a camera.

When light propagates through complex media, such as biological tissue, paint, clouds or even multimode fibers (MMFs), it is mixed into a high number of degrees of freedom leading to the observation of a seemingly random speckle pattern at the output[1,2], and limiting the information that can be transferred through them[3–5]. While the process leading to the generation of this intricate interference pattern is complex, owing to the deterministic and linear nature of the propagation of light in such media, the response of the system between a set of input and output modes is fully represented by a single matrix **H**. This matrix usually corresponds to the scattering matrix or part of it, such as the transmission or reflection matrices. While its derivation from analytical or numerical models is highly challenging and often impossible, experimentally, it can be measured via wavefront-shaping techniques[6,7]. This matrix gives us full knowledge over the wave propagation, thus enabling many applications in imaging[2,8–11], quantum information[12], among many others[1,2,13]. However, the dynamics of the system or external actions introduce perturbations into the known configuration, rendering our previous knowledge approximate at best. For applications in telecommunications and imaging, the detrimental effect of this perturbation can be bypassed by finding a set of channels that are insensitive to it[14–20]. When the changes depend on a single parameter $\zeta$ these channels can be identified as generalized principal modes which are insensitive to first-order variations of $\zeta$[15–17,19,21–23].

Nonetheless, in certain scenarios, the objective may shift from mitigating the impacts of perturbations towards actively enhancing them. This is the case, for example, when we want to use the output light for sensing applications[23–29], where it is possible to use wavefront shaping techniques to increase the interaction between the propagating light and the parameter of interest $\zeta$. The enhanced interaction causes the output field to become more sensitive to perturbations

[1]Institut Langevin, ESPCI Paris, Université PSL, CNRS, 75005 Paris, France. [2]Univ. Grenoble Alpes, CNRS, LIPhy, 38000 Grenoble, France.
✉e-mail: rodrigo.gutierrez-cuevas@espci.fr

induced by small changes of the parameter and to carry more information about it, usually quantified by the Fisher information $\mathcal{J}$. This increase in information allows for improved precision when estimating small changes in $\zeta$, according to the Cramér–Rao bound, which states that the variance of the estimation $\sigma_\zeta^2$ will be larger or equal to the reciprocal of the Fisher information, i.e. $\sigma_\zeta^2 \geq \mathcal{J}^{-1}$[30,31]. More general bounds have been derived for cases with limited resources, see e.g. refs. 32–34, but here we use the Cramér–Rao bound, which is easy to calculate and asymptotically reachable.

When one uses an external reference to access both the amplitude and phase of the output field, the channel maximizing the Fisher information carries almost all the information in its global phase[23]. This presents a significant constraint, since phase measurements in optics are highly susceptible to noise, and require a level of stability usually only available in laboratory conditions, making such an approach less suitable for real-life implementations. In comparison, protocols based on intensity measurements are quite robust and therefore broadly applicable[26–29]. In particular, the spatial information concealed within the speckle pattern of the light coming out of a complex medium has been exploited to develop a wide range of specklegram sensing devices[26–29]. However, the identification of the channel maximizing the information carried by its output intensity distribution, and the strat-

egy allowing to reach the precision limit remain unsolved. This is in part due to the fact that the relation between the input field and the output intensity distribution is not linear, but rather quadratic which prevents the use of standard matrix methods.

To solve these problems, we exploit the versatility of higher-order tensors to describe the quadratic relationship between the input field and its output spatial intensity distribution. We use this tensor-based framework to study three practical configurations for robust intensity-based sensing applications: first, when we only have control over the input wavefront for a fixed detection scheme at the distal end of the fiber; second, when controlling the projection of the outgoing field on the distal end for a fixed arbitrary input; and third, when controlling both the input wavefront and the output projections. In particular, we experimentally study the case of an MMF which is perturbed by pressing down on it transversally with a motorized actuator as exemplified in Fig. 1a. The parameter $\zeta$ represents the linear displacement of the actuator and $\mathbf{H}$ the transmission matrix (TM) between $M$ input modes formed by the 144 modes of the fiber (see Methods for more details), and a set of $N$ output modes, such as the pixels of a camera. With this implementation we are able to demonstrate an enhancement in the Fisher information by more than four orders of magnitude when using the optimal input-output configuration.

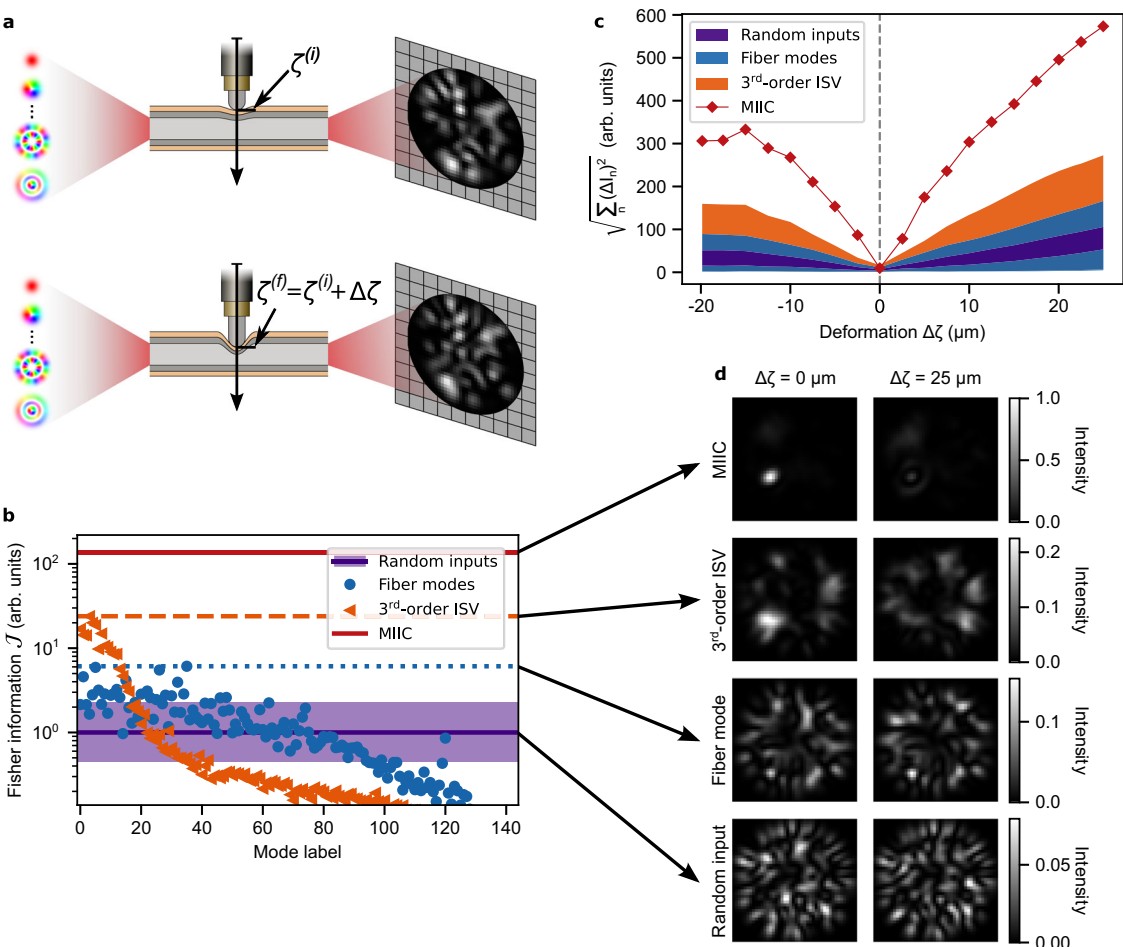

**Fig. 1 | Sensitivity to perturbations. a** Change on the output intensity distribution induced by a local deformation on a multimode fiber. **b** Fisher information in the pixel basis for random inputs (where the line denotes the mean value for one-thousand states and the shaded region the range obtained), the fiber modes (where the dotted line marks the maximum value), the third-order input singular vectors (ISVs) (where the dashed line marks the maximum value), and the maximum-

information intensity channel (MIIC). **c** Intensity change over large deformations for the same modes as in (**b**). For the random inputs, the fiber modes and the third-order ISVs the shaded region indicates the range of values. **d** Output intensity distribution for the MIIC, the most sensitive third-order ISV, the most sensitive fiber mode, as well as a random input for the reference deformation $\Delta\zeta = 0\ \mu m$ and a large deformation for which $\Delta\zeta = 25\ \mu m$.

## Results

### Optimizing the input field

Intuitively, the Fisher information carried by the output intensity distribution, under the assumption of Gaussian noise with a known standard deviation $\sigma$, is given by the change of the output intensity produced by first-order variations in $\zeta$,

$$\mathcal{J}(\zeta) = \frac{1}{\sigma^2}\sum_{i=1}^{N}\left(\partial_\zeta I_i\right)^2, \tag{1}$$

where $I_i = |e_i^{(\text{out})}|^2$ is the intensity measured at the $i^{\text{th}}$ output mode. Therefore, to determine the maximum-information intensity channel (MIIC), that is, the one maximizing the Fisher information, we need to find the input field $\mathbf{e}^{(\text{in})}$ that leads to the largest variations in output intensity distribution for small changes in the deformation $\zeta$. Note that the assumption of Gaussian noise is quite general encompassing all systems for which the noise fluctuations are dominated by dark and readout noise.

Given the nonlinear dependence of the Fisher information on the input field, one may think that there is no other choice than to cast this problem as a standard nonlinear optimization problem[18,35–38]. Nevertheless, rewriting the Fisher information in terms of the input field and TM as

$$\mathcal{J}(\zeta) = \frac{1}{\sigma^2}\sum_{i=1}^{N}\left(\sum_{jk}\mathcal{W}_{ijk}^{(3)*}e_j^{(\text{in})}e_k^{(\text{in})*}\right)^2, \tag{2}$$

allows codifying the nonlinear relation between the input field and the changes in the output intensity distribution induced by changes in $\zeta$ into the third-order tensor $\mathcal{W}^{(3)}$, defined component-wise as $\mathcal{W}_{ijk}^{(3)} = \partial_\zeta(H_{ij}^* H_{ik})$ where $H_{ij}$ are the components of the matrix $\mathbf{H}$. Furthermore, as demonstrated in Section 1 of the Supplementary Information, for a fixed set of output modes (such as the pixels of a camera) it is possible to rewrite the maximization of the Fisher information as a best rank-one approximation of $\mathcal{W}^{(3)}$[39,40],

$$\max_{\mathbf{e}^{(\text{in})}}\mathcal{J}(\zeta) = \frac{1}{\sigma^2}\max_{\mathbf{u},\mathbf{v}}\langle\mathcal{W}^{(3)},\mathbf{u}\otimes\mathbf{v}\otimes\mathbf{v}^*\rangle^2. \tag{3}$$

The right-hand side is given by the inner product between the third-order tensor $\mathcal{W}^{(3)}$, and the rank-one third-order tensor $\mathbf{u}\otimes\mathbf{v}\otimes\mathbf{v}^*$ with $\otimes$ denoting the outer product. Note that the optimizations are subject to the normalization constraints $\|\mathbf{e}^{(\text{in})}\| = \|\mathbf{u}\| = \|\mathbf{v}\| = 1$ which fix the total number of input photons. Therefore, in order to determine the MIIC we need to find the set of three vectors that best approximate $\mathcal{W}^{(3)}$, in the sense that they minimize the sum of the squared differences between their components.

Equation (3) shows that higher-order tensors can provide fresh perspectives in tackling intricate and nonlinear challenges[39–47]. Indeed, they allow drawing a clear parallel with optimizations of linear systems which can often be cast as best rank-one approximations of matrices whose solution is simply obtained by computing the singular-value decomposition (SVD) and taking the outer product of the first pair of singular vectors[15–17,19,20,23]. In nonlinear cases, however, we need to solve the equivalent problem for higher-order tensors which is not a simple task. Nonetheless, a step in the right direction can be taken by performing the higher-order singular value decomposition (HOSVD)[39,43,44]. This generalization of the matrix SVD allows decomposing a higher-order tensor in terms of a sum of rank-one tensors composed of higher-order singular vectors which form orthonormal bases for their

respective spaces. Specifically, for the third-order tensor $\mathcal{W}^{(3)}$ we have

$$\mathcal{W}^{(3)} = \sum_{i}^{N}\sum_{jk}^{M}\mathcal{S}_{ijk}^{(3)}\mathbf{u}_i^{(3)}\otimes\mathbf{v}_j^{(3)}\otimes\mathbf{v}_k^{(3)*} \tag{4}$$

in which the third-order singular vectors, $\mathbf{u}_i^{(3)}$ and $\mathbf{v}_j^{(3)}$, identify the main components of the $\mathcal{W}^{(3)}$, and $\mathcal{S}^{(3)}$ is known as the core tensor, which plays the role of the singular values for matrices and satisfies an ordering property that generally arranges the third-order singular vectors from the most to the least relevant (see Section 2 of the Supplementary Information and Ref. 39,43,44 for more details). Therefore, the first third-order input singular vectors (ISVs), $\mathbf{v}_k^{(3)}$, correspond to input fields that are highly sensitive to changes in $\zeta$. Moreover, one big advantage of the HOSVD is that it can be computed in a straight forward manner in terms of SVDs of different rearrangements of the tensor into matrices[39,43,44].

Hence, to find the third-order ISVs of $\mathcal{W}^{(3)}$ we first need to construct this tensor. This is done by performing two measurements of $\mathbf{H}$, using the pixels of a camera as output modes, around a reference value $\zeta = \zeta^{(i)}$ for the deformation. With these two measurements $\mathcal{W}^{(3)}$ can be constructed by approximating the derivative with respect to $\zeta$ via finite differences (see Methods for more details about the experimental implementation). Then, its HOSVD can be computed.

In Fig. 1b we compare the values of the Fisher information obtained for the third-order ISVs with those obtained when using the fiber modes and one-thousand random wavefronts as inputs. It can be seen that the Fisher information for the first third-order ISVs is above all the values obtained with the fiber modes and random wavefronts, and with the maximum value being an order of magnitude larger the maximum value attained for a random wavefront. Therefore, the HOSVD immediately provides us with a set of modes generally ordered from the one with the largest Fisher information to the smallest, and with the first ones being highly sensitive to the perturbation. Note, however, that the ordering is not strict, which is a well-known feature of the HOSVD. This difference with the matrix SVD stems from the choice that needs to be made for higher-order tensors between a diagonal or orthogonal decomposition, as it is generally not possible to have both[39,43,44].

One of the main differences between the HOSVD and the SVD is that the first singular vectors do not immediately provide us with the best rank-one approximation. Nonetheless, they form a reasonably good first guess that can be used as a seed for a modified version of the iterative alternating-least squares algorithm[39,40] to determine the best rank-one approximation. In this algorithm, we iterate over the vectors forming the best-rank one approximation and solve the least-squares problem resulting from leaving the other vectors fixed. This iterative procedure is simple to implement and relies solely on standard matrix operations (see Methods for details). This allows us to identify the MIIC which provides a two order of magnitude boost in Fisher information with respect to an average random wavefront, as shown in Fig. 1b.

It is also worth noting that, as seen in Fig. 1c, both the third-order ISVs and the MIIC maintain their high sensitivity over a large range of deformations. Likewise, in Section 5 of the Supplementary Information, we show that they also keep their sensitivity if the deformation is slightly displaced along the fiber's length. However, by looking at the intensity distributions in Fig. 1d, it can be seen that their behavior is quite different. For the most sensitive third-order ISV, we get an intensity distribution that is fairly distributed throughout the pixels of the camera. Under the effect of the deformation, this intensity is redistributed towards neighboring pixels thus keeping a fairly distributed pattern. In contrast, the intensity distribution of the MIIC is mostly localized around a focal spot. This optimal distribution allows concentrating the maximum amount of information across a few pixels in order to increase the signal-to-noise ratio (SNR). The main effect of the deformation on this distribution is to dim the focal spot by

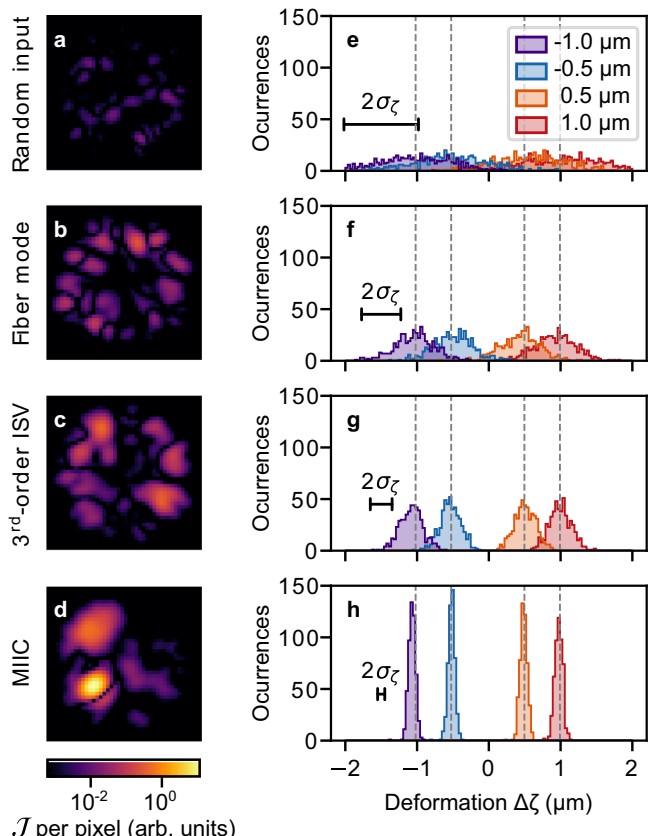

**Fig. 2 | Estimating a change in the perturbation. a–d** Measured Fisher information per output pixel (all plots share the same scale bar). **e–h** Histograms of estimated changes in the deformation for four different values ($\Delta\zeta = -1$, $-0.5$, $0.5$, $1\,\mu m$) marked with dashed gray lines and with the mean width $2\sigma_\zeta$ marked in black. Both types of plots are shown for a random input (**a**, **e**), the most sensitive fiber mode (**b**, **f**), the optimal third-order input singular vectors (ISV) (**c**, **g**), and the maximum-information intensity channel (MIIC) (**d**, **h**).

redistributing the energy to other areas. This local change in intensity of the focal spot can be recorded by a single detector thus minimizing the cost of the system and allowing high-speed operations. Despite the similarity, the MIIC is not equivalent to the channel obtained through phase conjugation to focus at the same spot[7] and for which the Fisher information value is much lower.

A direct application of the MIIC regards the estimation of changes in the perturbation $\Delta\zeta$. Assuming $\Delta\zeta$ to be small, when compared to the deformation that decorrelates the output, we can safely assume a linear model for which we only need to perform calibration measurements for the reference intensity distribution and its derivative with respect to $\zeta$ above the noise level. Figure 2a–d show the Fisher information per output pixel obtained from these calibration measurements for four different input fields, namely, a random wavefront, the fiber mode with the highest Fisher information, the third-order ISV with the highest Fisher information and the MIIC. Here, the drastic enhancement provided by the MIIC can be appreciated.

For the actual estimation, we use the same four input fields, but perform the measurements at a lower input power and for the four smaller deformation values $\Delta\zeta = -1$, $-0.5$, $0.5$, and $1\,\mu m$. The input intensity is lowered for two reasons; first, to place ourselves in a situation dominated by Gaussian noise, and second, to reduce the SNR in order to clearly appreciate the difference when estimating the deformations using the different input fields. We measure 500 intensity distributions for each field and each deformation, and use each one of these measurements to estimate the deformation using

the corresponding minimum variance unbiased estimator[30] (see Methods for more details). The results are shown in Fig. 2e–h. For the random input, it is practically impossible to discriminate the four peaks corresponding to the four different deformation values. For the fiber mode and the third-order ISVs, we can clearly discern the deformations that are further apart as the distributions are narrower. However, there is still significant overlap between the neighboring ones. Finally, when we take a look at the results for the MIIC, we can appreciate four well-defined peaks with a standard deviation that is more than an order of magnitude smaller than the one obtained for the random input, in agreement with the Cramér-Rao lower bound.

## Optimizing the output projection modes

One consequence of the quadratic relationship between the input field and the measured output intensity is that the Fisher information is not invariant under changes in the output projection modes (OPMs). While the pixels of a camera might be the simplest output modes to implement experimentally, they are generally not the optimal choice since they are blind to the information that could be hidden in the relative phase variations of the field from one pixel to another, and they spread out the information across many modes, thus decreasing the SNR. To address both of these issues we could foresee demultiplexing the output field into a specifically designed set of spatial channels, which is experimentally feasible using Fourier filters[48–50], photonic lanterns or multiplane light converters[51–53]. This is the principle behind techniques that allow increasing the information transfer for telecommunication applications[54,55], estimating the three-dimensional position and orientation of single molecules[48–50], or beating Rayleigh's curse when imaging two closely-spaced sources[56–59], among others[60,61].

Mathematically, this spatial demultiplexing is performed by projecting the output field onto an orthogonal set of $Q$ OPMs, where $Q$ can be smaller than the number of output modes used to define **H**. If we assume that the input field is fixed, then, as shown in Section 1 of the Supplementary Information, the Fisher information can be rewritten as

$$\mathcal{J}(\zeta) = \frac{1}{\sigma^2} \sum_{q=1}^{Q} \langle \mathbf{p}_q, \mathbf{E}_\zeta \cdot \mathbf{p}_q \rangle^2, \qquad (5)$$

where $\mathbf{p}_q$ is the $q^{th}$ OPM, and $\mathbf{E}_\zeta = \partial_\zeta(\mathbf{e}^{(out)} \otimes \mathbf{e}^{(out)*})$ is a rank-2 Hermitian matrix. Therefore, the Fisher information can be maximized by choosing as OPMs the two eigenvectors of $\mathbf{E}_\zeta$ with nonzero eigenvalues (see Section 3 of the Supplementary Information for the proof and the explicit expressions of the eigenvectors). These two optimal OPMs are given by a simple linear combination of the output field $\mathbf{e}^{(out)}$ and its derivative with respect to the parameter $\partial_\zeta \mathbf{e}^{(out)}$. For unitary systems, the OPMs are given by the symmetric and antisymmetric combinations of the output field $\mathbf{e}^{(in)}$ and the orthogonal component of the derivative $\partial_\zeta \mathbf{e}^{(in)}$. This result is a generalization of those previously derived for estimating the distance between two particles where the output field is projected onto a Gaussian and a first-order Hermite-Gauss modes, which resemble the symmetric and antisymmetric superpositions of the two-point spread functions, respectively[56–58,61].

Figure 3 shows the impact of projecting onto the optimal OPMs on the Fisher information for the same random inputs and fiber modes as those used in Fig. 1b. By using the optimal OPMs, the mean Fisher information obtained when using random wavefronts as inputs is an order of magnitude larger than the maximum value obtained with the MIIC in the pixel basis, which clearly shows the benefits of choosing the OPMs appropriately even when one cannot control the input wavefront.

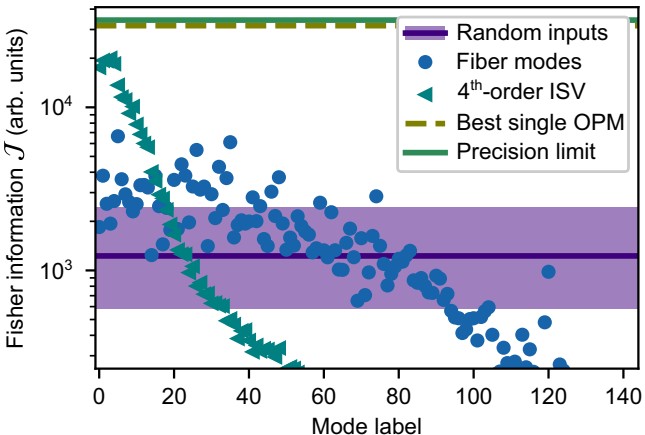

**Fig. 3 | Adapting the output projections modes.** Effect of using the optimal output projection modes (OPMs) on the Fisher information per photon for various input fields. It shows the Fisher information obtained when using the optimal OPMs for each of the random inputs used in Fig. 1, the fiber modes, and the fourth-order input singular vectors (ISVs). Also shown are the Fisher information values attained for the optimal input-output combination, which defines the precision limit for intensity-based measurements, and for the optimal channel when a single OPM is used.

## Optimal input-output combination: reaching the precision limit

Let us now find the optimal channel when we are free to shape the input field and choose the OPMs. This optimal input-output combination is the channel that maximizes the Fisher information when projected onto its two optimal OPMs, and sets the precision limit achievable with intensity measurements. To find it, we rewrite the Fisher information in terms of just two OPMs and the input field which leads to,

$$\mathcal{J}(\zeta) = \frac{1}{\sigma^2} \sum_{q=1}^{2} \left\langle \mathcal{W}^{(4)}, \mathbf{p}_q^* \otimes \mathbf{p}_q \otimes \mathbf{e}^{(in)} \otimes \mathbf{e}^{(in)*} \right\rangle^2, \tag{6}$$

where we defined the fourth-order tensor $\mathcal{W}^{(4)}$ whose components are given by $\mathcal{W}^{(4)}_{ijkl} = \partial_\zeta (H_{ik}^* H_{jl})$. Even though this expression resembles the one in Eq. (3), it does not correspond to a rank-two approximation of $\mathcal{W}^{(4)}$. Nonetheless, we can still use tensor-based techniques to obtain an excellent first guess by computing the HOSVD of $\mathcal{W}^{(4)}$,

$$\mathcal{W}^{(4)} = \sum_{ij}^{N} \sum_{kl}^{M} \mathcal{S}^{(4)}_{ijkl} \mathbf{u}^{(4)*}_i \mathbf{u}^{(4)}_j \otimes \mathbf{v}^{(4)}_k \otimes \mathbf{v}^{(4)*}_l. \tag{7}$$

Figure 3 shows the resulting Fisher information for the corresponding fourth-order ISVs, $\mathbf{v}_k^{(4)}$, when projected onto their respective optimal OPMs. Once more, these higher-order ISVs provide us with a generally ordered orthogonal basis of highly-sensitive modes, with the first ones surpassing all the modes of the fibers and random inputs.

To reach the precision limit, however, we need to perform a nonlinear optimization using the first fourth-order ISV as a seed (see Methods for details). The results shown in Fig. 3 demonstrate that the Fisher information achieved by this channel is well over two orders of magnitude above that obtained for the MIIC that uses the pixels as OPMs. Nevertheless, this maximum value is very close to that provided by the first few fourth-order ISVs. Given the freedom to adapt the OPMs to the output field, only the information contained in the global phase of the output field is lost. Therefore, it is this quantity that dictates the difference in precision of the solution achieving the precision limit for intensity measurements with respect to the one dictated by the quantum Cramér-Rao bound[23]. However, the global phase

is the quantity that is most likely to be corrupted by noise due to the mechanical instabilities limiting its usefulness for real-life applications.

Another approach that would simplify the experimental implementation of the spatial demultiplexing is to use a single OPM to monitor change of intensity at the output. In this case we have a single term in Eq. (6) so that finding the optimal combination reduces to finding the best rank-one approximation of $\mathcal{W}^{(4)}$. As can be seen in Fig. 3, this simpler approach almost allows us to reach the precision limit, for which the Fisher information value is only 4.8% larger. For practical applications, it means that one can approach the precision limit within a very small margin using a single photodetector. In fact, the output fields produced by the first fourth-order ISV and the best rank-one approximation are highly similar to the one reaching the precision limit, with which they have field correlations that are above 78%. This similarity can also be appreciated in their intensity profiles, shown in Fig. 4. More surprising, however, is that the first third-order ISV obtained with the output pixel basis is also highly similar to the one reaching the precision limit (see Fig. 4). This is not intuitive since $\mathcal{W}^{(3)}$ does not contain any information about the phase at the output, which $\mathcal{W}^{(4)}$ does have. These similarities also demonstrate that the HOSVD allows getting really close to the precision limit while bypassing the need for a nonlinear optimization.

## Discussion

By introducing tensor-based methods to the study of complex systems, we have provided a natural framework for studying intensity-based measurements. Here, we used lower-rank approximations of higher-order tensors to find the channels that are most sensitive to a given perturbation, i.e. that suffer the largest change in output intensity when the perturbation changes. Their high sensitivity allows them to be used for highly robust and precise sensing applications which we demonstrated experimentally by estimating small perturbations in an MMF. It was also shown that what is meant by most sensitive is highly dependent on the choice of output modes used to measure the intensity distribution. This dependence was exploited to find the channel that allows extracting the most information available at the output, and thus achieve the precision limit of intensity-based measurements.

The tensor-based framework introduced here provides several fundamental and practical advantages. It allows drawing a clear parallel between linear and nonlinear optimizations problems in complex systems which can be written as lower-rank approximations of matrices and higher-order tensors, respectively. Likewise, it places the optimization problem within the appropriate mathematical context which is particularly important here since it allows benefitting from advances of a branch of mathematics devoted to the study of lower-rank approximations of higher-order tensors. Lastly, the tensor-based approach allows using the HOSVD to find highly sensitive intensity-based channels, while bypassing the need for a nonlinear optimization. This excellent first guess can then be refined with iterative algorithms specifically designed to find lower-rank approximations.

This framework opens the door to further investigations, such as the development of highly sensitive distributed specklegram MMF sensors in which deformations can be sensed throughout the whole length of the fiber. In Section 6.1 of the Supplementary Information, we show how the Fisher information evolves as the position of the deformation along the fiber changes from the one used to find the optimal modes. These results show that, while the configuration allowing reaching the precision limit remains highly sensitive for a significant range, for longer variations of the position it exhibits high variations in sensitivity. In contrast, the third-order ISVs with the pixels as OPMs generalize better to other locations. Therefore, it should be possible to find a field that optimizes the sensitivity to a deformation regardless of its location. Likewise, we show that the tensor framework provides us with another way to approach this problem. We can

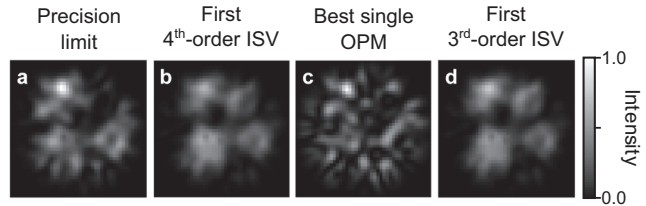

Precision limit | First 4ᵗʰ-order ISV | Best single OPM | First 3ʳᵈ-order ISV

**Fig. 4 | Output fields near the precision limit.** Output intensity distribution for the mode achieving the precision limit (**a**), the first fourth-order input singular vectors (ISV) (**b**), the best mode when using a single output projection mode (OPM) (**c**), and the first third-order ISV (**d**) (all plots share the same scale bar).

instead find the OPMs that provide the best precision on average for any given input field. As shown in Section 6.2 of the Supplementary Information, an excellent candidate is provided by the output fourth-order singular vectors, $\mathbf{u}^{(4)}$ given by the HOSVD of $\mathcal{W}^{(4)}$. Additionally, the tensor-based relations established here can also be used for the study of incoherent systems where a TM cannot be derived. This last point could help gain further insight into fluorescent imaging applications through scattering media. It can also be expected for the fields presented here to find other applications, such as for focusing light inside scattering media but with properties that will be quite different compared to using the generalized principal modes[17].

## Methods

### Experimental setup
The optical setup is represented in Fig. S1 of the Supplementary Information. The light source consists of a continuous linearly polarized laser beam at 1550 nm (TeraXion NLL) injected into a 10:90 polarization-maintaining fiber coupler (PNH1550R2F1). The 90% arm illuminates a digital micromirror device (DMD) (Vialux V-650L) which modulates the input field. The light is converted into left circular polarization using a quarter-wave plate. The shaped field is then left-circularly polarized and imaged with a 4f system onto the input facet of a 25 cm-long step-index fiber with a 50 $\mu$m core and 0.22 numerical aperture. The output facet is imaged via another 4f system onto an InGaAs camera (Xenics Cheetah 640-CL 400 Hz) after passing through a quarter-wave plate, followed by a beam displacer to select the left-circularly polarized component. The other 10% arm is used to produce a tilted reference that is made to interfere with the signal field in order to retrieve the output field via off-axis holography[62]. A shutter allows blocking the reference field to perform intensity measurements of the signal field. The fiber can be deformed by pressing on it using a servo motor actuator (Thorlabs Z812).

### TM and $\mathcal{W}^{(3)}$ measurements
Using Lee holograms to shape the amplitude and phase of the input light with the DMD[63–65], we first measure the TM in the pixel basis. This is achieved by sending 7200 square layouts consisting of $37 \times 37$ square macropixels whose value is either zero or a random phase of amplitude one. The corresponding output fields are recovered from the interferograms between the reference and signal fields, and subsequently projected onto a square pattern of $44 \times 44$ macropixels formed by grouping $4 \times 4$ pixels of the camera. Regrouping all input and output fields into the columns of matrices $\mathbf{X}$ and $\mathbf{Y}$, respectively, we reconstruct the TM via $\mathbf{H} = \mathbf{Y} \cdot \mathbf{X}^{-1}$ where $\mathbf{X}^{-1}$ denotes the pseudoinverse of $\mathbf{X}$. The 144 fiber modes are identified by computing the SVD of the resulting TM and taking all the singular vectors that have close to unit singular values (see Fig. S2 of the Supplementary Information). All subsequent TM measurements are performed by sending 1440 random inputs obtained by randomly superimposing all 144 fiber modes. To determine the third-order tensor $\mathcal{W}^{(3)}$, we

measure two TMs, $\mathbf{H}^{(\pm)}$, for two different values of the deformation, $\zeta^{(i)} \pm d\zeta/2$, centered around the reference value $\zeta^{(i)}$ and with $d\zeta = 3$ $\mu$m. These two measurements are then used to approximate the derivative with respect to $\zeta$ using finite differences. We use a similar approach to construct $\mathcal{W}^{(4)}$.

### Optimizing the Fisher information
A detailed explanation about the different optimizations for each case can be found in Secs. 2 and 3 of the Supplementary Information. Here, we provide a summary of the main steps taken for each case.

To find the MIIC, given by the best rank-one approximation of $\mathcal{W}^{(3)}$, the first step is to compute the third-order singular vectors. These are given by the left singular vectors of the matrices obtained by choosing the $n^{\text{th}}$ index of the tensor to be the rows and all the other indices are arranged to form the columns. This is known as the mode-$n$ matricization[39,43]. For example, if $n = 2$ then we get the $M \times MN$ matrix $\mathbf{W}^{(3)}$ defined component-wise via the following index assignment, $W_{IJ}^{(3)} = \mathcal{W}_{ijk}^{(3)}$ where $I = j$ and $J = i + (k-1)N$. The left singular vectors of $\mathbf{W}^{(3)}$ are the third-order ISVs. The next step is to use as a seed the rank-one tensor formed by all the first higher-order singular vectors for an iterative alternating-least squares (ALS) algorithm to solve the following minimization problem

$$\min_{\mathbf{a},\mathbf{b},\mathbf{c}} \| \mathcal{W}^{(3)} - \mathbf{a} \otimes \mathbf{b} \otimes \mathbf{c} \|. \tag{8}$$

For the ALS algorithm, instead of tackling the minimization over $\mathbf{a}$, $\mathbf{b}$, and $\mathbf{c}$ all at once, we iterate over each vector by solving the standard least-square problem that results from leaving the other ones fixed. For example, we start by solving

$$\min_{\mathbf{a}} \| \mathcal{W}^{(3)} - \mathbf{a} \otimes \mathbf{b} \otimes \mathbf{c} \|, \tag{9}$$

for which we can find the exact solution. Then, we do the same for $\mathbf{b}$ and $\mathbf{c}$. This is repeated for a predefined number of times or until a convergence criterion is satisfied. If this standard implementation fails to provide a solution satisfying the symmetries of the original tensor, then it is possible to adapt the ALS algorithm by simply imposing the symmetries at the end of each loop (see Section 2 of the supplementary information).

When we only seek to optimize the OPMs for a fixed input field, it suffices to construct the rank-two Hermitian matrix $\mathbf{E}_\zeta = \partial_\zeta (\mathbf{e}^{(\text{out})} \otimes \mathbf{e}^{(\text{out})*})$ and find its two eigenvectors with nonzero eigenvalues. The explicit expressions are given in Section 3 of the Supplementary Information. However, if we seek to optimize both the input field and the OPMs then we need another approach. In the case in which we restrict ourselves to a single OPM, then we only need to find the best rank-one approximation of the fourth-order tensor $\mathcal{W}^{(4)}$. Therefore, in this case we can proceed as we did to find the MIIC. For the more general case of finding the optimal input field and corresponding OPMs, we need to perform a nonlinear optimization over the input field. However, just like for the ALS algorithm, we use the first fourth-order ISVs of $\mathcal{W}^{(4)}$ as an initial guess.

### Experimental estimation of the deformation
Before performing the experimental estimation of small deformations, we verified the Gaussian noise assumption by performing several measurements of the output fields for the four fields used for the estimation for two input intensity values. These measurements confirmed that at low intensity the Gaussian assumption was indeed valid (see Section 5.1 of the Supplementary Information for details). Given that the perturbations under consideration are small, we use a linear model for the change in intensity distribution as a function of $\Delta\zeta$. The measured intensity distribution $\chi$ over the output modes is

then given by

$$\chi \approx \mathbf{I}(\zeta^{(i)}) + \partial_\zeta \mathbf{I}(\zeta^{(i)})\Delta\zeta + \boldsymbol{w}(\sigma), \tag{10}$$

where $\mathbf{I}(\zeta^{(i)})$ represents the output intensity distribution over the output modes prior to changing the deformation, $\partial_\zeta \mathbf{I}(\zeta^{(i)})$ is the derivative evaluated at $\zeta^{(i)}$, and $\boldsymbol{w}(\sigma)$ is a vector representing the Gaussian noise with zero mean and standard deviation $\sigma$. For this linear model the minimum variance unbiased estimator[30] is given by

$$\Delta\zeta^{(\text{est})}(\boldsymbol{\chi}) = \frac{\partial_\zeta \mathbf{I}(\zeta^{(i)}) \cdot \left[\boldsymbol{\chi} - \mathbf{I}(\zeta^{(i)})\right]}{\parallel \partial_\zeta \mathbf{I}(\zeta^{(i)}) \parallel^2}. \tag{11}$$

Both the reference intensity distribution $\mathbf{I}$ and the derivative $\partial_\zeta \mathbf{I}$ are calibrated using independent measurements (see Section 5.2 of the Supplementary Information for further details). This estimator is used to obtain all the estimations of $\Delta\zeta$ shown in Fig. 2.

## Data availability
The data generated in this study have been deposited in the dedicated GitHub repository[66].

## Code availability
All the code used to produce the results and figures presented in this work can be found in the dedicated GitHub repository[66].

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

## Acknowledgements
R.G.C. acknowledges L.L. Sánchez-Soto and K. Liang for useful discussions. The authors acknowledge the French Agence Nationale de la Recherche (Grant No. ANR-20-CE24-0016 MUPHTA and No. ANR-23-CE42-0010-01 MUFFIN, S.M.P.) and the Labex WIFI (ANR-10-LABX-24, ANR-10-IDEX-0001-02 PSL*, S.M.P.).

## Author contributions
R.G.C. conceived the project, developed the theory and performed the experimental measurements. S.M.P. supervised the experimental work and theory development. D.B and J.d.R. contributed to the scientific discussion and interpretation of the results. All authors contributed to the writing of the manuscript.

## Competing interests
The authors declare no competing interests.

## Additional information

**Peer review information** : *Nature Communications* thanks the anonymous reviewer(s) for their contribution to the peer review of this work. A peer review file is available.

