## [Peer Review File · Nature Communications]

Reaching the precision limit with tensor-based wavefront shapingREVIEWER COMMENTS

Reviewer #1 (Remarks to the Author):

In this article the authors present a tensor based formalism of random scattering which allows them to find what is termed an "optimal crashing channel", which in short is an input and detection mode that maximises the sensitivity of the output signal to a specific perturbation. This is illustrated numerically and experimentally for a deformation of a multimode fiber. This is an interesting work and will garner a good level of interest from the community, but I feel there are a number of necessary revisions before it is suitable for publication. These are detailed below.

1. The authors state that the tensor framework they propose is necessary since the measurand, optical intensity, is quadratic in the scattering/transmission matrix. Whilst it does seem to give a convenient framework in which to find optimal results, there have been a good number of results in the literature that have maximised Fisher information (FI) from measured intensity distributions without the need for such a 'complicated' framework. For example Moerner has engineered optimal PSFs localization measurements of single emitters (<https://doi.org/10.1103/PhysRevLett.113.133902> and associated follow ups). Further discussion of existing works in the literature and the context of the current work is needed to justify their approach.
2. In a similar vein a number of works have been reported discussing information limits in random channels which should be cited (e.g. <https://doi.org/10.1038/nphys4036>, <https://doi.org/10.1103/PhysRevE.67.036621>, <https://doi.org/10.1088/1367-2630/aba063>)
3. L97 - in what sense do the authors mean "best"?
4. L107 - The authors mention they take two measurements of H. After having read through the article and the supplement a number of times I am still unclear as to what these two measurements are and why multiple are necessary.
5. In a similar vein - a lot of details are missed in the main text and the reader is thus

required to read the supplementary detail (which for the most part is helpful) to actually fully understand the article. The previous point is one example, but others include suddenly introducing different rank approximations/representations, inference of deformation and so on. This is not the purpose of the supplemental material and the article should be understandable on its own. Clear and fuller discussions are required given the highly technical nature of material. If the length of Nat Comms does not allow for this, then the article is more suited for a different journal.

6. L114 - Maybe this is a gap in my knowledge, but what do the authors mean by generally ordered? They say the ordering is from the largest FI, but Fig 1b do not seem to show that the first mode has the largest FI.

7. The authors claim they are "crashing into disorder". This nomenclature is then used to justify the definition of an "optimal crashing channel". Frankly, I see in no way that the authors are "crashing into disorder". In fact this phrase and terminology means nothing and provides no physical insight. It strikes me as sensationalist and 'click bait' and has no place in an objective scientific article. The authors are not the first to note that the effects of disorder can be 'optimised away', nor does the disorder actually help them achieve greater sensitivity than a non-disordered system (at least this is not shown/claimed), so how do the authors imagine they have crashed into anything? I would very strongly urge use of a meaningful phrase and revision of the title.

8. The optimal input and output identified by the authors, which gives rise to greater sensitivity than use of a random input/pixel output basis (which would not be expected to be optimal in the first place), is designed for a very specific deformation. In the authors example this is a deformation of a fiber at a specific position along its length. How does the sensitivity drop off as the deformation differs from the perturbation used in design. How does FI decay with lateral displacement of the deformation along the fiber length? Does the gain in sensitivity for the designed deformation come at the cost of a loss of sensitivity to any other type of deformation? This is an important aspect relating to whether such a scheme has practical use in sensor design. In sensing systems, the perturbation is often not so well controlled e.g.

deformation of a fiber could be anywhere along its length, temperature fluctuations are random distributed and binding position in biosensors is likewise random. I thus have concerns as to the actual practicality of using the 'optimal crashing channel'.

9. In the supplement, The definition of 'an' in Eq 8/7b seems somewhat circular at the moment (given the $J^{1/2}$ factor). Furthermore on L342 it's stated $||a|| = J^{1/2}$, whilst on 347 its stated $||a|| = 1$. Which is it? It's also unclear why there is the freedom to normalise a in the first place since it derives directly from H (a fixed system parameter), sigma (noise - again fixed) and e_{in} (the input field which is already normalised, so you cant normalise it further). There is no freedom here. Given these mathematical operations and definition of an is quite central to their formalism, greater explanation needs to be given to each step that they perform.

10. I am also further confused by the fact that in moving to Eq 10 (in the supplement) u, which has now replaced the vector a in their expressions is again treated as a degree of freedom over which they can optimise. Of course the vector a does contain e_{in} (and thus has some freedom over which to optimise), the functional form and dependence of a on H (through W) does not seem to be incorporated into the minimisation as a constraint. Put another way, u should depend on v and w in the optimisation so as to match the constraints embodied in the form of Eq 7c.

11. L422/423. What is the core tensor S? Also is there a type in Eq 17 where a lower case s is used?

12. L491 - the authors claim that they do not see intensity dependent noise at lower intensities and this confirms the assumption of Gaussian noise that they have made. Intensity independent noise does not mean that the noise is Gaussian. Indeed one can have Gaussian distributed noise which is signal dependent (e.g. at high intensity shot noise which is Poisson distributed is very well approximated by a Gaussian distribution, but it is still intensity dependent). Thus this claim is unfounded. To support their claims the authors need to (at minimum) give a p-value of whether the statistical distribution of their noise is explainable by a Gaussian distribution.

13. Eq 22 - The I_{ζ} notation for the derivative is quite hard to read and distinguish from $I(\zeta)$. I'd suggest a more readable notation.

14. Below Eq 23 - The authors 'calibrate their derivative' by setting 'the input power as high as possible'. By the previous analysis this takes them into the regime of signal dependent noise which is not considered. Discussion of why this is valid is needed or at least a discussion of the approximation made and possible error that can result.

15. The complexity of the calibration and optimisation procedure may pose a limit to use of the technique. Can the authors comment on potential simplifications that they might expect to be possible in the future.

In summary, the topic of this article is interesting and could potentially be published in Nat Comm, but before it can be there are a number of fundamental aspects that need to be addressed to convince me of practicality, validity, clarity and hence to the potential impact.

Reviewer #2 (Remarks to the Author):

The goal of this paper is to identify the optimal channel that maximizes the change in its output intensity distribution and the Fisher information encoded in it about a given perturbation in complex systems. To this end, the authors use third-order tensors to find corrections to the Fisher information. They have demonstrated the method experimentally, by estimating small perturbations in a multimode fiber, underlining the robustness of the approach.

I think the paper contains new and interesting results that deserve to be published.. However, I have some major concerns that should be addressed before I can make my final decision.

- Higher-order tensors is an appropriate tool to deal with nonlinear (sometimes quantum) effects. However, we know from the time-honored example of $su(2)$ that these tensors should transform properly under the symmetry of the problem. In this way, one gets the

famous multipole expansion that gives us rotationally invariant results. Are the authors sure about this property in this case? Because if not, a change in, say, the reference frame would modify the tensors in a nontrivial way.

- In the end, the third-order tensor used by the authors enters as a modification of the Fisher information. In the frequentist approach, such modifications are well known. As such, they lead to two famous bounds, Barankin and Bhattacharya, that take into account higher order corrections to the likelihood. There is a huge literature in the field that the authors seem to ignore. I think they should check that and compare with their method.

Actually, this is especially relevant for limited resources (that is, the number of trials), where the Cramer-Rao bound is only attainable as an asymptotic limits and those corrections play a major role.

- The authors concentrate only on intensity detection. However, I am sure one might try to compute the quantum Fisher information, which has the advantage of being dependent exclusively on the state. It would be nice to see how far are the results of this paper from the ultimate quantum limits.

- Concerning the optimal projecting modes, the previous experience with demultiplexing methods suggest that those modes are, essentially, derivatives of the point-spread-function (PSF). How this translates into the present scenario.

I am convinced that these questions are relevant to improve the readability of the paper.

Authors' response to reviews of **Reaching the precision limit with tensor-based wavefront shaping**

R. Gutiérrez-Cuevas, D. Bouchet, J. de Rosny, and S.M. Popoff

7 May 2024

We would like to thank all the reviewers for the time they have spent reviewing our work and for their insightful comments. In what follows we address all their comments/concerns and indicate the corresponding modifications made to the manuscript.

Reviewer 1

Reviewer comment

In this article the authors present a tensor based formalism of random scattering which allows them to find what is termed an "optimal crashing channel", which in short is an input and detection mode that maximises the sensitivity of the output signal to a specific perturbation. This is illustrated numerically and experimentally for a deformation of a multimode fiber. This is an interesting work and will garner a good level of interest from the community, but I feel there are a number of necessary revisions before it is suitable for publication. These are detailed below.

Authors' response

We are glad the reviewer enjoyed our work, and thank them for their thorough review. In the following we detail how each of their comments have been addressed.

Reviewer comment

1. The authors state that the tensor framework they propose is necessary since the measurand, optical intensity, is quadratic in the scattering/TM. Whilst it does seem to give a convenient framework in which to find optimal results, there have been a good number of results in the literature that have maximised Fisher information (FI) from measured intensity distributions without the need for such a 'complicated' framework. For example Moerner has engineered optimal PSFs localization measurements of single emitters (<https://doi.org/10.1103/PhysRevLett.113.133902> and associated follow ups). Further discussion of existing works in the literature and the context of the current work is needed to justify their approach.

Authors' response

As the reviewer rightly points out, there have been other works that have maximized the Fisher information for intensity measurements in other scenarios. Indeed, for any given system, once the correct expression for the Fisher information is known, one can always use a standard non-linear optimization algorithm to maximize its value. This is the approach performed in

prior works on this topic. However, due to the non-convex nature of the problem and the high-dimension of the solution space, by starting from an arbitrary configuration it is very unlikely to converge to the ideal solution. Moreover, without an expression for the ideal solution, it is difficult to quantify the efficiency of the solution compared to the optimal one. Finally, it obscures any connection that can exist with other physical situations.

In contrast, the tensor formalism that we are introducing here addresses all of these challenges. Specifically, it provides us with the following fundamental and practical advantages which we elaborate in more detail in what follows:

1. Draws a clear parallel between linear and nonlinear problems in complex systems.
2. Places the optimization within an insightful mathematical context.
3. Provides an excellent first guess through the higher-order singular value decomposition.
4. Provides specifically designed algorithms for further refinement.
5. Paves the way for more general scenarios in which the location of the perturbation is not fixed.
6. The formulation in terms of the transmission matrix (TM) provides a more general approach that is directly applicable to a wide range of applications.

Framing this nonlinear optimization problem as a lower-rank tensor approximation comes with various advantages, both fundamental and practical. (1.) Firstly, it draws a clear parallel between linear problems which are commonly written in terms of lower-rank approximations of matrices. Examples of these are finding modes enhancing the range of memory effects, or those mitigating nefarious effects such as dispersion. Therefore, it demonstrates that similar nonlinear problems can be incorporated into a more general framework given in terms of lower-rank approximations of higher-order tensors. (2.) Likewise, our approach places this type of nonlinear optimization within the appropriate mathematical context. This is particularly important here since there is a whole branch of mathematics dedicated to the study of lower-rank approximations of higher-order tensors. (3.) Additionally, the tensor-based approach also comes with computational advantages. Through the use of the higher-order singular value decomposition (HOSVD), we are able to find a set of modes that almost achieve the precision limit (see Fig. 3 of the manuscript) while bypassing the need for a nonlinear optimization since the HOSVD is computed via the singular value decomposition of rearrangements of the tensor into matrices. Moreover, this higher-order singular vector provides an excellent first guess for further refinement in order to reach the precision limit. (4.) This refinement can then be carried out with iterative algorithms specifically designed for finding lower-rank approximations. These algorithms rely solely on standard matrix operations which makes them simpler to implement than standard nonlinear optimizations algorithms. (5.) Lastly, from the point 8 made by the reviewer asking how our approach extends to more general scenarios, we were able to further show the power of the tensor-based approach. Specifically, we show in our response to said point that the higher-order vectors allow finding input fields and output projection modes (OPMs) that are highly sensitive even if the perturbation changes location.

In the current manuscript we mentioned the case of estimating the distance from two closely space emitters where it was shown that by changing the measurement basis it was possible to

beat Rayleigh's curse. However, as it is being pointed out, optimizing the Fourier window for estimating the three-dimensional location and orientation of dipolar emitters is another example that we now mention in our current work. One of the difference between these two situations and our work, however, is that they both consider as the input field the light coming out of the point emitters. Therefore, there is no possibility to tailor the input field since it is interlinked with the parameters of interest, and the Fisher information can only be optimized by adapting the detection of the output field which can be seen as a change of basis. Likewise, we now mention methods for single molecule tracking that use structured illumination to enhance the localization precision. However, in this latter case the input field is only optimized using a small set of parameters thus significantly restricting the solution space.

(6.) In comparison, in our work we consider the complete scenario in which both the input field and the output projection basis can be adapted in order to extract the maximum amount of information. Moreover, by using the TM formalism, we show that our results can be applied to any type of linear media, complex or not. Likewise, the use of the TM inherently takes into account the discrete and finite nature of the input and output degrees of freedom use to generate and measure optical fields. It is these general considerations that lead to a tensor-based framework for finding the optimal modes for estimating small changes in a perturbation. Our formalism could then, in turn, be applied to enhance the previously mentioned works. Specifically, for the estimation of the three-dimensional location and orientation of dipolar emitters or the separation of two point sources, we could incorporate the structuring of the illumination, whereas for the single molecule tracking we could adapt the OPMs instead of simply focusing the output light into photodetectors. This should provide a boost in precision that has not been exploited.

Modifications

We have revised the manuscript in order to better justify the tensor-based approach and showcase its advantages compared to the state of the art. In particular, we have added the following paragraph in the Outlook section that clearly states the benefits of our tensor-based approach.

The tensor-based framework introduced here provides several fundamental and practical advantages. It allows drawing a clear parallel between linear and nonlinear optimizations problems in complex systems which can be written as lower-rank approximations of matrices and higher-order tensors, respectively. Likewise, it places the optimization problem within the appropriate mathematical context which is particularly important here since there is a whole branch of mathematics that is devoted to the study of lower-rank approximations of higher-order tensors. Lastly, the tensor-based allows the use of the HOSVD to find highly sensitive fields, while bypassing the need for a nonlinear optimization. Moreover, if this excellent first guess is to be refined, it also provides us with iterative algorithms specifically designed to find lower-rank approximations.

Likewise, we have added Refs. [36,37] for the structuring of light for tracking single molecules and Refs. [47-49] for the estimation of the three-dimensional location and orientation of dipolar emitters.

Reviewer comment

2. In a similar vein a number of works have been reported discussing information limits in random

channels which should be cited (e.g. <https://doi.org/10.1038/nphys4036>,
<https://doi.org/10.1103/PhysRevE.67.036621>, <https://doi.org/10.1088/1367-2630/aba063>)

Authors' response

While these works use different metrics to quantify the information flux which are not adapted for the task of parameter estimation, we agree with the reviewer that they should be mentioned as part of this work, and we thank them for pointing them out.

Modifications

We have added Refs. [3-5] in the first paragraph of the introduction.

Reviewer comment

3. L97 - in what sense do the authors mean "best"?

Authors' response

The best rank-one approximation is defined as the rank-one tensor which minimizes the Frobenius distance to the original tensor, that is, it minimizes the Frobenius norm of the tensor obtained by taking the element-wise difference between the two. Explicitly, for a third-order tensor \mathcal{T} , the best rank-one approximation is the solution to the minimization problem:

$$\min_{\mathbf{a}, \mathbf{b}, \mathbf{c}} \|\mathcal{T} - \mathbf{a} \otimes \mathbf{b} \otimes \mathbf{c}\|^2 = \min_{\mathbf{a}, \mathbf{b}, \mathbf{c}} \sum_{i=1}^I \sum_{j=1}^J \sum_{k=1}^K |\mathcal{T}_{ijk} - a_i b_j c_k|^2. \quad (1)$$

However, in our work we make use of the fact that the solution to this problem is equivalent to the following maximization problem:

$$\max_{\tilde{\mathbf{a}}, \tilde{\mathbf{b}}, \tilde{\mathbf{c}}} |\langle \mathcal{T}, \tilde{\mathbf{a}} \otimes \tilde{\mathbf{b}} \otimes \tilde{\mathbf{c}} \rangle|^2 \text{ subject to } \|\tilde{\mathbf{a}}\| = \|\tilde{\mathbf{b}}\| = \|\tilde{\mathbf{c}}\| = 1. \quad (2)$$

Within the particular context of our work, finding the best rank-one approximation is equivalent to maximizing the Fisher information. This connection has been clarified in the revised version (see our response to points 9 and 10). Moreover, following the reviewer's comment (point 5) about the technical nature of our work, we have decided to explicitly state the definition of the best rank-one approximation in the main text, and to add more details in the Supplementary Information.

Modifications

We have added a clarifying sentence at the end of the paragraph after Eq. (3).

Therefore, in order to determine the MIIC we need to find the set of three vectors that best approximate $\mathcal{W}^{(3)}$, in the sense that they minimize the sum of squared differences between their component.

A detailed description of the best rank-one approximation problem is now provided in Sec. 2.1 of the Supplementary information.

Reviewer comment

4. L107 - The authors mention they take two measurements of H . After having read through the article and the supplement a number of times I am still unclear as to what these two measurements are and why multiple are necessary.

Authors' response

The third-order tensor $\mathcal{W}^{(3)}$ is defined in terms of a derivative with respect to the parameter ζ of the product between the components of the TM via

$$\mathcal{W}_{ijk}^{(3)} = \partial_{\zeta}(\mathbf{H}_{ij}^* \mathbf{H}_{ik}). \quad (3)$$

Therefore, measurements of the \mathbf{H} for at least two different values around a reference value of ζ are needed to compute the derivative via finite differences. In the previous version this was not explained explicitly, and we thank the reviewer for pointing it out. In the current version, this has been made explicit in the main text, and a more detailed explanation is provided in the Methods and Supplementary Information.

Modifications

We have added the following clarifying sentence in the paragraph introducing the HOSVD.

With these two measurements $\mathcal{W}^{(3)}$ can be constructed by approximating the derivative with respect to ζ via finite differences (see Methods for more details about the experimental implementation).

More details have also been included in the Methods and Supplementary Information.

Reviewer comment

5. In a similar vein - a lot of details are missed in the main text and the reader is thus required to read the supplementary detail (which for the most part is helpful) to actually fully understand the article. The previous point is one example, but others include suddenly introducing different rank approximations/representations, inference of deformation and so on. This is not the purpose of the supplemental material and the article should be understandable on its own. Clear and fuller discussions are required given the highly technical nature of material. If the length of Nat Comms does not allow for this, then the article is more suited for a different journal.

Authors' response

We agree with the reviewer that, in an attempt to synthesize the essential parts of our work, we might have made it harder for interested readers to fully understand some of the concepts that are being used for the first time within this context. Therefore, we have decided to restructure the more technical parts of our work by bringing some of them into the main manuscript, mainly those dealing with the key concepts of the tensor-based approach. The corresponding information contained in the Methods has been further synthesized to keep only the essential information with fuller and in depth discussions now being contained in the Supplementary Information. Likewise, we have moved the detailed proofs for the Fisher information expressions from the Methods to the Supplementary Information.

Modifications

Specifically, we have made the following modifications of the main manuscript:

1. Introduced the Fisher information in terms of intensity changes (Eq. (1)), and explicitly showed how by using the TM the changes of intensity can be described by a third-order tensor (Eq. (2)).
2. Provided more details about the connection between finding the best rank-one approximation and the maximization of the Fisher information.
3. Given the explicit form of the HOSVD, and more context about it (see the paragraph surrounding Eq. (4) and Eq. (7)).
4. Added some clarification for the estimation process.
5. Mentioned explicitly how $\mathcal{W}^{(3)}$ and $\mathcal{W}^{(4)}$ are constructed.

We now include a Supplementary Information document in which we provide a detailed presentation of all the proofs and in depth presentation of the various concepts and experimental implementation. The Methods have been synthesized to provide only the essential information for the main results.

Reviewer comment

6. L114 - Maybe this is a gap in my knowledge, but what do the authors mean by generally ordered? They say the ordering is from the largest FI, but Fig 1b do not seem to show that the first mode has the largest FI.

Authors' response

Generally ordered means that the values are ordered on averaged, with the first ones having a higher probability of providing a larger value for the Fisher information. This in contrast with the 2D matrix SVD which always gives access to the singular values sorted by decreasing order. The difference stems from a choice that needs to be made when decomposing higher-order tensors into rank-one components. We need to choose between a diagonal decomposition, that is with a fully-diagonal core tensor, and orthogonal singular vectors, it is generally not possible to have both (see Refs. [39,40,44]). The HOSVD chooses the latter, and with the loss of diagonality of the core tensor there is no unique way to order the different components. The particular choice that is made leads to Fisher information values that are only ordered on average. We further explain this point the main text.

Modifications

We have added the following sentences to the main text to further explain this point.

Note, however, that the ordering is not strict which is a well-known feature of the HOSVD. This difference with the matrix SVD stems from the choice that needs to be made for higher-order tensors between a diagonal or orthogonal decomposition, it is generally not possible to have both [38,42,43].

Reviewer comment

7. The authors claim they are "crashing into disorder". This nomenclature is then used to justify the definition of an "optimal crashing channel". Frankly, I see in no way that the authors are "crashing into disorder". In fact this phrase and terminology means nothing and provides no physical insight. It strikes me as sensationalist and 'click bait' and has no place in an objective scientific article. The authors are not the first to note that the effects of disorder can be 'optimised away', nor does the disorder actually help them achieve greater sensitivity than a non-disordered system (at least this is not shown/claimed), so how do the authors imagine they have crashed into anything? I would very strongly urge use of a meaningful phrase and revision of the title.

Authors' response

The title comes from the fact that the channels that lead to the largest change in output intensity do so by concentrating light around the perturbation. Therefore, we can see them as crashing straight into the disorder induced by the perturbation which is in contrast to the Wigner-Smith principal modes which can be used to avoid it. Nonetheless, we understand the reviewer's comment and have thus removed "Crashing into disorder" from the title for the sake of clarity and redefined the optimal crashing channel as the "maximum intensity information channel" (MIIC).

Modifications

The title has been changed to "Reaching the precision limit with tensor-based wavefront shaping" and renamed the channel maximizing the Fisher information as the maximum intensity information channel (MIIC).

Reviewer comment

8. The optimal input and output identified by the authors, which gives rise to greater sensitivity than use of a random input/pixel output basis (which would not be expected to be optimal in the first place), is designed for a very specific deformation. In the authors example this is a deformation of a fiber at a specific position along its length. How does the sensitivity drop off as the deformation differs from the perturbation differs from that used in design. How does FI decay with lateral displacement of the deformation along the fiber length? Does the gain in sensitivity for the designed for deformation come at the cost of a loss of sensitivity to any other type of deformation? This is an important aspect relating to whether such a scheme has practical use in sensor design. In sensing systems, the perturbation is often not so well controlled e.g. deformation of a fiber could be anywhere along its length, temperature fluctuations are random distributed and binding position in biosensors is likewise random. I thus have concerns as to the actual practicality of using the 'optimal crashing channel'.

Authors' response

The main goal of this work is to provide new tensor-based tools for dealing with intensity-based measurements and to determine their precision limits for intensity based measurements. Determining this limit is essential in order to assess which sensing parameters can be measured by a given sensor and with which precision. However, the reviewer raises an important and interesting point that made us perform some supplementary calculations based on the data acquired with some very interesting results which further demonstrate the suitability of the tensor-based approach.

To sum up the additional work done, that we detail in the following, we first present our model to simulate realistically the effect of the displacing the location of the deformation. Two cases are distinguished: for specific sensors, one may want the system to be sensitive to a given location and nowhere else, and for distributed sensors, keeping a high sensitivity for all positions is preferable. We numerically study both cases using measured data and present the results. We show that our approach is beneficial in both cases, extending the range of applications of our approach.

Let us begin by considering what happens to the Fisher information for each of the fields considered in this work when the location of the deformation changes. From Fig. 1 in the main text it could already be appreciated that the heightened sensitivity of the MIIC and the third-order input singular vectors remains even when the strength of the deformation changes. However, to study what happens when the location of the deformation changes, we consider the following model. Given that the deformation is applied at a particular point along the fiber, we can divide our system, and thus the TM, into three part:

$$\mathbf{H} = \mathbf{H}_2 \cdot \mathbf{D} \cdot \mathbf{H}_1, \quad (4)$$

where \mathbf{H}_1 represents the transmission of light up to the region that is deformed, \mathbf{D} represents the propagation through the perturbation, and \mathbf{H}_2 the propagation through the final stretch following the deformation. Propagation through small stretches of an unperturbed fiber of length Δz can be modelled by the diagonal matrix, $\exp(i\beta\Delta z)$, where β is the diagonal matrix containing the propagation constants of the modes of the fiber. Therefore, one way to model the displacement of the deformation is by applying the operator representing the axial shifts to reduce the propagation distance on one side and increase it on the other side

$$\mathbf{H}(\Delta z) = \mathbf{H}_2 \cdot \exp(i\beta\Delta z) \cdot \mathbf{D} \cdot \exp(-i\beta\Delta z) \cdot \mathbf{H}_1. \quad (5)$$

Given that the length of the fiber is fairly small and the deformation not too strong, the matrices \mathbf{H}_2 and \mathbf{H}_1 are mainly diagonal matrices (see e.g. Refs. [10,19]). Therefore, it is safe to assume that \mathbf{H}_2 and \mathbf{H}_1 commute with $\exp(i\beta\Delta z)$ thus giving

$$\mathbf{H}(\Delta z) = \exp(i\beta\Delta z) \cdot \mathbf{H}_2 \cdot \mathbf{D} \cdot \mathbf{H}_1 \cdot \exp(-i\beta\Delta z). \quad (6)$$

With this model we can compute the Fisher information for the modes found for Δz at various locations along the fiber's length. The results are shown in Fig. 1. These plots show that the optimal fields found at Δz remain quite sensitive within a range of 0.5mm. Depending on the application, this can be more than enough to account for inaccuracies during its implementation.

It is important to make a distinction between different use cases. The results provided in our work are perfectly suited to enhance the precision of point sensors. This type of sensor is designed for a specific parameter and at a specific location, which is what we consider. In this case, a significant drop in the Fisher information as we move away from the region of interest would be beneficial for these sensors since it would make them less sensitive to perturbations happening in other parts of the system.

If, instead, the goal is to develop a distributed sensor which can detect perturbations with high sensitivity along the full length of the fiber then the solution will be different. Nonetheless, our formalism can help address this problem as well. From Fig. 1 it can be seen that the third-order ISV actually provides us with a channel that remains more sensitive than the fiber modes

Figure 1 | Changing the location of the perturbation. **a-b**, Evolution of the Fisher information with respect to the location of the perturbation for one-thousand random inputs (continuous line show the mean and the shaded region the full range), the maximum value of the fiber modes, the maximum value of the third-order ISVs, the MIIC, the channel using a single output projection mode, and the channel allowing reaching the precision limit by tailoring both the input and output. Note that for each value Δ the mean value for the random inputs has been used to rescale the others to simplify the comparison. Two different ranges for Δz are shown **a**, a large range of 2cm (**a**) and a smaller range of 2mm (**b**).

even for large variations of the position of the deformation. This shows that it should be possible to find an input field that is optimal for sensing perturbations irrespective of their location and that the HOSVD provides us again with an excellent starting point, showing again the usefulness of the techniques we are introducing.

Along the same lines, it can be seen that the channels for which the OPMs have been optimized exhibit large variations in their Fisher information as a consequence of their specificity to the deformation and specific output field. Much less noticeable variations are present for the channels using the pixel basis as OPMs because this basis is not optimized for any field in particular. These features point to another strategy for dealing with distributed perturbations in which optimal OPMs that do not depend specifically on the input field need to be found. Here again, our formalism shows us the way. Using the fourth-order output singular vectors (OSVs) of the fourth-order tensor $\mathcal{W}^{(4)}$ as OPMs, it is possible to boost the precision for any input field on average. Figure 2 shows the histograms and mean values obtained when computing the Fisher information for one thousand random inputs when the OPMs are the pixels of a camera, the modes of the fiber, and the fourth-order OSVs.

The benefits of using other OPMs compared to the pixels of the camera become obvious. Even when just 10 of the fiber modes or the fourth-order OSVs are used, they significantly increase the mean value with respect to that obtained with the pixel basis. In particular, the fourth-order OSVs exhibit a much higher mean Fisher information than the one obtained with the fiber modes, and remains a better candidate even when only two of its elements are used as OPMs. Similarly to what we did before, it is also possible to study their behavior as we change the position of the deformation. The results are shown in Fig. 3 where it can be seen that while there is still a clear dependence on the specific location of the deformation, the fourth-order OSVs provide a better alternative to the fiber modes and the pixels of a camera throughout a large displacement range of the perturbation.

Given the importance of these results we have decided to add them to the Supplementary Information file describing the model used to change the location of the perturbation, the results

Figure 2 | Blind estimation. a-c, Histogram and mean value (vertical lines) of the Fisher information for 1000 random inputs when projecting the output field onto the pixel basis, the fiber modes, and the output higher-order singular vector. The number of modes used at the output for the fiber modes and the fourth-order OSVs is a, 30, b, 10, and c, 2.

Figure 3 | Blind estimation with changing location. Mean value of the Fisher information obtained for one-thousand random input fields using as OPMs the pixels of a camera, the fiber modes and the fourth-order OSVs. For the pixel basis the shaded region indicates the range obtained and for the fiber modes and the fourth-order OSVs three different curves are shown for each, the continuous line uses 30 elements, the dashed line uses 10, and the dotted line only 5.

and the corresponding discussion. These results are mentioned in the concluding remarks of the main text.

Another point that was raised by the reviewer is the inclusion of other types of perturbations. In this regard, it is not possible to say how the fields found in our work would perform without the corresponding measurements or a model. Nonetheless, it is possible to consider this multi-parameter case by extending our formalism to this more general case. However, given the large number of results already being presented and the risk in confusing the reader we feel that this extension falls out of the scope of this work.

Modifications

The results just presented have been added to the Supplementary Information in Sec. 6. Likewise, we have added the following paragraph in the Outlooks of the main manuscript.

This new framework opens the door to further investigations, such as the development of highly sensitive distributed specklegram MMF sensors in which deformations can be sensed throughout the whole length of the fiber. In Sec. 6.1 of the Supplementary Information, we show how the Fisher information evolves as the position of the deformation along the fiber changes from the one used to find the optimal modes. These results show that, while the configuration allowing reaching the precision limit remains highly sensitive for a significant range, for longer variations of the position it exhibits high variations in sensitivity. In contrast, the third-order ISVs with the pixels as OPMs generalize better to other locations. Therefore, it should be possible to find a field that optimizes the sensitivity to a deformation regardless of its location. Likewise, we show that the tensor framework provides us with another way to approach this problem. We can instead find the OPMs that provide the best precision on average for any given input field. As shown in Sec. 6.2 of the Supplementary Information, an excellent candidate is provided by the output fourth-order singular vectors, $\mathbf{u}^{(4)}$ given by the HOSVD of $\mathbf{W}^{(4)}$.

Reviewer comment

9. In the supplement, The definition of 'an' in Eq 8/7b seems somewhat circular at the moment (given the $J^1/2$ factor). Furthermore on L342 it's stated $\|a\| = J^{1/2}$, whilst on 347 its stated $\|a\| = 1$. Which is it? It's also unclear why there is the freedom to normalise a in the first place since it derives directly from H (a fixed system parameter), σ (noise - again fixed) and e_i (the input field which is already normalised, so you cant normalise it further). There is no freedom here. Given these mathematical operations and definition of a is quite central to their formalism, greater explanation needs to be given to each step that they perform.

10. I am also further confused by the fact that in moving to Eq 10 (in the supplement) u , which has now replaced the vector a in their expressions is again treated as a degree of freedom over which they can optimise. Of course the vector a does contain e_i (and thus has some freedom over which to optimise), the functional form and dependence of a on H (through W) does not seem to be incorporated into the minimisation as a constraint. Put another way, u should depend on v and w in the optimisation so as to match the constraints embodied in the form of Eq 7c.

Authors' response

We thank the reviewer for pointing out these issues. In the current version we have revised the proof, and made the connection to the best rank-one approximation clearer. In short, the Fisher

information can be written as the inner product of the third-order tensor $\mathcal{W}^{(3)}$ and a rank-one tensor according to

$$\mathcal{J}(\zeta) = \frac{1}{\sigma^2} \left\langle \mathcal{W}^{(3)}, \mathbf{a} \otimes \mathbf{e}^{(\text{in})} \otimes \mathbf{e}^{(\text{in})*} \right\rangle^2, \quad (7)$$

where the vector \mathbf{a} is defined component-wise as

$$a_n = \frac{1}{\sigma \mathcal{J}^{1/2}(\zeta)} \sum_{m,m'}^M \mathcal{W}_{nmm'} e_m^{(\text{in})} e_{m'}^{(\text{in})*}. \quad (8)$$

Note that $\|\mathbf{a}\| = \|\mathbf{e}^{(\text{in})}\| = 1$, that is, all the vectors forming the rank-one approximation are normalized. As pointed out by the reviewer, the vector \mathbf{a} depends on \mathbf{H} and the input field therefore one would think that this dependence needs to be incorporated into the optimization, regardless of the algorithm used. However, as is now shown in the revised proof, if we consider instead the cost function

$$C = \left\langle \mathcal{W}^{(3)}, \mathbf{u} \otimes \mathbf{v} \otimes \mathbf{v}^* \right\rangle, \quad (9)$$

where $\|\mathbf{u}\| = \|\mathbf{v}\| = 1$, and \mathbf{u} is independent of \mathbf{v} . This cost function has more degrees of freedom than \mathcal{J} , and is easy to show that $C = \sigma \mathcal{J}^{1/2}$ if

$$u_n = \frac{\sum_{mm'} \mathcal{W}_{nmm'} v_m^* v_{m'}}{\sum_n (\sum_{mm'} \mathcal{W}_{nmm'} v_m^* v_{m'})^2}, \quad (10)$$

for $\mathbf{e}^{(\text{in})} = \mathbf{v}$. Now, if we find \mathbf{u} and \mathbf{v} such that C is a maximum then it is easy to verify that \mathbf{u} is related to \mathbf{v} according to Eq. (10) and $C = \sigma \mathcal{J}^{1/2}$. Moreover, since \mathcal{J} lives in a smaller subspace than C , at these points \mathcal{J} is also at a maxima for $\mathbf{e}^{(\text{in})} = \mathbf{v}$. This demonstrates that the input field that maximizes the Fisher information can be found by computing the best rank-one approximation of $\mathcal{W}^{(3)}$.

We hope that this cleaner proof helps elucidate the connection with the rank-one approximation and avoid any future confusion.

Modifications

The revised proof is presented in Sec. 1.2 of the Supplementary Information.

Reviewer comment

11. L422/423. What is the core tensor S ? Also is there a type in Eq 17 where a lower case s is used?

Authors' response

The core tensor is the equivalent of the singular values for the higher-order singular value decomposition. This tensor satisfies an ordering and orthogonality properties that, in the matrix case, reduce to the diagonality of the matrix containing the singular values and their ordering. For higher-order tensors, however, it is generally not possible to decompose them as the outer product of orthogonal vectors and have a fully-diagonal core tensor as we mentioned to our response to point 6.

Figure 4 | Noise QQ plot. Quantile-quantile plot for the measured intensity deviation from its mean value including every pixel, realization and measured field when compared to a Gaussian distribution for high and low input intensity values.

Modifications

We now briefly explain the role of the core tensor and its properties just after the introduction of the HOSVD in Eq. (4). Likewise we provide a detailed summary of the main properties in Sec. 2.2 of the Supplementary Information.

The lower-case typo has also been fixed.

Reviewer comment

12. L491 - the authors claim that they do not see intensity dependent noise at lower intensities and this confirms the assumption of Gaussian noise that they have made. Intensity independent noise does not mean that the noise is Gaussian. Indeed one can have Gaussian distributed noise which is signal dependent (e.g. at high intensity shot noise which is Poisson distributed is very well approximated by a Gaussian distribution, but it is still intensity dependent). Thus this claim is unfounded. To support their claims the authors need to (at minimum) give a p -value of whether the statistical distribution of their noise is explainable by a Gaussian distribution.

Authors' response

The reviewer is right to point this out, intensity independent noise does not entail that it follows a Gaussian distribution. Nonetheless, our data does follow the assumed distribution at low intensity values. In the revised version, we have added a Figure in the Supplementary Information showing the quantile-quantile plot for the measured intensity deviation from its mean value including every pixel, realization and measured field when compared to a Gaussian distribution for high and low intensity values.

This plot, shown here in Fig. 4 for convenience, compares the cumulative distribution functions obtained from the theoretical Gaussian distribution and the data. If the data follows the model then the points obtained (shown as blue dots) should follow a straight line at 45° (shown as a continuous red line). For the high intensity data, we see a clear deviation for positive values where the points clearly take larger values than those predicted by a Gaussian distribution. This means that the distribution for the data has a positive heavy tail which is consistent with

the intensity dependent noise that we see in Extended Data Figure 3. However, this deviation disappears at lower intensity values where all the points follow closely the predicted diagonal line, thus demonstrating that our assumption of Gaussian noise is valid.

Modifications

Figure 4 has been added in Sec. 5.1 of the Supplementary Information along with the corresponding discussion.

Reviewer comment

13. Eq 22 - The I_ζ notation for the derivative is quite hard to read and distinguish from $I(\zeta)$. I'd suggest a more readable notation.

Authors' response

The notation for the derivative has been changed in order to avoid any possible confusion.

Modifications

We have changed the derivative notation from I_ζ to $\partial_\zeta I$ to avoid confusion.

Reviewer comment

14. Below Eq 23 - The authors 'calibrate their derivative' by setting 'the input power as high as possible'. By the previous analysis this takes them into the regime of signal dependent noise which is not considered. Discussion of why this is valid is needed or at least a discussion of the approximation made and possible error that can result.

Authors' response

When performing the calibration measurements we simply want to place ourselves in a situation of high signal-to-noise ratio regardless of the distribution followed by the noise. In our case, the increase in noise due to the intensity dependence is almost negligible with respect to the increase in the corresponding signal. This can be appreciated in the Fig. S3 in the Supplementary Information. Figure 5, shown here, further supports this point by showing more accurate estimations are obtained when using the calibration at higher intensity than the equivalent performed at low intensity. For this reason, it is a valid approach to perform the calibration at higher intensity values since it yields more accurate results.

Reviewer comment

15. The complexity of the calibration and optimisation procedure may pose a limit to use of the technique. Can the authors comment on potential simplifications that they might expect to be possible in the future.

Authors' response

This is a good point being raised and that we did not clearly explain in the original manuscript. The complex calibration needing measurements at high and low intensities is only necessary because we wanted to isolate the effects of the inherent sensibility of the input field and those

Figure 5 | Estimation with varying intensity calibration. Histograms of estimated changes in the deformation for four different values ($\Delta\zeta = -1, -0.5, 0.5, 1\ \mu\text{m}$) marked with dashed gray lines and with the mean width $2\sigma_\zeta$ marked in black. These histograms are shown when calibrating the derivative at high intensity value (left column) and at a low intensity value (right column). The results are shown for a random input (first row), the most sensitive fiber mode (second row), the optimal third-order ISV (third row), and the MIIC (fourth row). Note that the mean width of the histograms is not shown for the random input with the calibration at low intensity since it is larger than the range showed in the plot.

arising from a poor calibration. Given that the random input (the worst performing field) generates a small change in the output intensity distribution, it was not possible to accurately estimate the derivative at a low intensity. As can be seen from the Fig. S5 in the Supplementary Information, even the mean intensity difference remains quite noisy. However, this is not the case for the MIIC. Figure 5 compares what happens when the estimation is performed with the derivative estimated using the high or low intensity measurements. What becomes apparent from these results is that the estimation of the random input is highly degraded due to the poor calibration of the derivative. We see a similar behavior for the fiber mode and the third-order ISV, but to a lesser degree. However, for the MIIC the estimation remains practically unchanged. Therefore, when using our result this complex calibration is not necessary. Another direction for future refinement would be to use an event-based camera that directly records changes in the intensity distribution, thus bypassing the need to record a reference image.

Modifications

We give now provide a justification for the calibration at high intensity and why it was needed for our work in Sec. 5.2 of the Supplementary Information. Likewise, we have added the following sentence explaining why the input intensity was lowered.

The input intensity is lowered for two reasons; first, to place ourselves in a situation dominated by Gaussian noise, and second, to reduce the SNR in order to clearly appreciate the difference when estimating the deformations using the different input fields.

Reviewer comment

In summary, the topic of this article is interesting and could potentially be published in Nat Comm, but before it can be there are a number of fundamental aspects that need to be addressed to convince me of practicality, validity, clarity and hence to the potential impact.

Authors' response

We hope to have addressed all the reviewers points in sufficient detail.

Reviewer 2

Reviewer comment

The goal of this paper is to identify the optimal channel that maximizes the change in its output intensity distribution and the Fisher information encoded in it about a given perturbation in complex systems. To this end, the authors use third-order tensors to find corrections to the Fisher information. They have demonstrated the method experimentally, by estimating small perturbations in a multi-mode fiber, underlining the robustness of the approach. I think the paper contains new and interesting results that deserve to be published.. However, I have some major concerns that should be addressed before I can make my final decision.

Authors' response

We thank the reviewer for their kind comments. In what follows we address each of their comments.

Reviewer comment

- Higher-order tensors is an appropriate tool to deal with nonlinear (sometimes quantum) effects. However, we know from the time-honored example of $su(2)$ that these tensors should transform properly under the symmetry of the problem. In this way, one gets the famous multipole expansion that gives us rotationally invariant results. Are the authors sure about this property in this case? Because if not, a change in, say, the reference frame would modify the tensors in a nontrivial way.

Authors' response

This is an interesting point that we did not want to detail at the risk of confusing readers and because it does not directly impact our results. We do not assume any particular symmetry structure, thus our results can be applied to ordered and fully disordered systems.

Nonetheless, we do consider changes of bases for which we need to transform the relevant tensors. As the reviewer points out, it is not hard to see that the third order tensor $\mathbf{W}^{(3)}$ does not transform in the usual way. Let \mathbf{U} be a unitary transformation representing a symmetry of our system, then the TM follows the following equation

$$\mathbf{H} = \mathbf{U}^\dagger \mathbf{H} \mathbf{U}. \quad (11)$$

This relationship indicates that our system is invariant with respect to \mathbf{U} , this is, for example, the case of physical rotation for multimode fibers with a cylindrical core. Now, if we try to apply

the same principle to our tensor we get the following

$$\mathcal{W}_{ijk}^{(3)} = \sum_{lmnp} U_{il}^T U_{in}^\dagger H_{lm}^* H_{np} U_{mj}^* U_{pk}, \quad (12)$$

where \mathbf{U}^T and \mathbf{U}^\dagger denote the transpose and conjugate transpose of \mathbf{U} . The modification induced by the transformation is not immediately clear unless the link with the TM is made explicit, which means that we cannot identify the tensor we are transforming in the right-hand side. This non-standard transformation rule stems from the fact that the output space corresponds to a different physical quantity than the input space. Given a specific input basis and OPMs, this transformation basis plays no role in our work and one obtains the results that we present in the first part of our work. However, when the OPMs are allowed to be tailored this non-standard transformation is what makes the Fisher information depend on the OPMs, and allows to further boost the precision of our system.

This odd transformation also stems from the fact that $\mathcal{W}^{(3)}$ is made up of a subpart of a fourth-order tensor that does transform as expected, namely $\mathcal{W}_{ijkl}^{(4)} = \partial_\zeta(H_{ik}^* H_{jl})$, through $\mathcal{W}_{ijk}^{(3)} = \mathcal{W}_{iijk}^{(4)}$. This fourth-order tensor is the one we use to fully optimize our system by tailoring the input field and OPMs. In comparison, the transformation for this fourth-order tensor is given by

$$\mathcal{W}_{ijkl}^{(4)} = \sum_{mnpq} U_{im}^T U_{jn}^\dagger \mathcal{W}_{mnpq}^{(4)} U_{pk}^* U_{ql}, \quad (13)$$

which is the standard transformation.

While these mathematical relations are interesting in their own right, they play no direct role in our current work. Moreover, the higher-order tensors introduced here simply provide us with a multidimensional array with which we can codify the nonlinear link between the input field and the output intensity, similarly to what is done in data science applications. But given their physical significance it is certainly something that we would be interested in studying further in the future.

Modifications

We have added a mention to this effect in Sec. 1.4 of the Supplementary Information.

Reviewer comment

- In the end, the third-order tensor used by the authors enters as a modification of the Fisher information. In the frequentist approach, such modifications are well known. As such, they lead to two famous bounds, Barankin and Bhattacharya, that take into account higher order corrections to the likelihood. There is a huge literature in the field that the authors seem to ignore. I think they should check that and compare with their method. Actually, this is especially relevant for limited resources (that is, the number of trials), where the Cramer-Rao bound is only attainable as an asymptotic limits and those corrections play a major role.

Authors' response

In our work we are not modifying the Fisher information, the third-order tensor appears from the quadratic relation linking the input field to the induced changes in the output intensity. Then,

we recast the optimization of our system in terms of lower-rank approximations of higher-order tensors. Since we are not considering the case of limited resources the Fisher information is well suited for our application. Moreover, the Fisher information is simpler to compute, not requiring higher-order derivatives or evaluating derivatives for various values of the parameter, and leads to a physically intuitive solution. Its pertinence is further confirmed by our experimental measurements where we get a clear boost in precision by appropriately choosing the input field. Nonetheless, we do think they deserve to be mentioned, and so a comment has been added to this effect in the main text.

Modifications

We have added the following sentence to our introduction.

More general bounds have been derived for cases with limited resources, see e.g. Refs. [32-34], but here we use the Cramér-Rao bound, which is easy to calculate and asymptotically reachable.

Reviewer comment

- The authors concentrate only on intensity detection. However, I am sure one might try to compute the quantum Fisher information, which has the advantage of being dependent exclusively on the state. It would be nice to see how far are the results of this paper from the ultimate quantum limits.

Authors' response

As we briefly mention in the introduction, the case of measuring the full field was treated in Ref. [23]. There it was shown that for field measurements, the optimal field places almost the entirety of the information in its global phase. The problem with this approach is that it requires the use of a stable reference for the interferometric measurement. This is a significant constraint for sensing systems, requiring a stability that can only be found in research laboratories, and for which small temperature fluctuations can hamper the measurement. Because of these issues we were motivated to study intensity measurements as they show better stability, and usually are easier to implement, thus making them better suited for real life applications.

Nonetheless, by allowing the OPMs to be adapted to the particular field, we are effectively performing field measurements with the caveat that the global phase is lost. Therefore, since the only information that we are losing is the one contained in global phase then it is this quantity that dictates the difference in precision between our solution and the one attaining the quantum limit. Exactly how far they are from each other will depend on the specific system under study and the type perturbation. In our particular case we cannot make a direct comparison with the quantum Cramer-Rao bound since we do not have direct access to the number of photons detected by the camera due to its calibration.

We have included a sentence detailing this trade off in the main text.

Modifications

We have added the following sentences at the end of the paragraph following Eq. (7).

Given the freedom to adapt the OPMs to the output field, only the information contained in the global phase of the output field is lost. Therefore, it is this quantity that dictates the difference in precision of the solution achieving the precision limit for intensity measurements with respect to the one dictated by the quantum Cramér-Rao bound [23]. However, the global phase is the quantity that is most likely to be corrupted by noise due the mechanical instabilities limiting its usefulness for real-life applications.

Reviewer comment

- Concerning the optimal projecting modes, the previous experience with demultiplexing methods suggest that those modes are, essentially, derivatives of the point-spread-function(PSF). How this translates into the present scenario.

Authors' response

In the previous version we mentioned that the optimal OPMs were simply given by some linear combination of the output field itself and its derivative with respect to the parameter. However, it is possible to solve for these modes explicitly, and we have included the derivation in Sec. 1.4 of the Supplementary Information. There, we show that for systems in which the perturbation does not induce losses, the optimal OPMs are given by the symmetric and antisymmetric combinations of $\mathbf{e}^{(\text{out})}$ and the orthogonal component of the derivative $\partial_{\zeta}\mathbf{e}^{(\text{out})}$. This is in essence the same result found for the two point source problem as shown in Ref. [43]. There, the output field is composed of two PSFs that, when sufficiently close together, they look like a single PSF. The derivative with respect to their separation then takes the form of the derivative of a PSF. In this sense, our results allow identifying the OPMs for more general problems.

Modifications

We have added the following sentence after Eq. (5) further explaining this point

For unitary systems, the OPMs are given by the symmetric and antisymmetric combinations of the output field $\mathbf{e}^{(\text{in})}$ and the orthogonal component of the derivative $\partial_{\zeta}\mathbf{e}^{(\text{in})}$. This result is a generalization of those previously derived for estimating the distance between two particles where the output field is projected onto a Gaussian and a first-order Hermite-Gauss modes, which resemble the symmetric and antisymmetric superpositions of the two point spread functions, respectively [56-58].

Likewise, we have added Sec. 1.4 to the Supplementary Information with the full derivation of the optimal OPMs in the general case.

Reviewer comment

I am convinced that these questions are relevant to improve the readability of the paper.

Authors' response

We hope to have address all the reviewer's concerns with sufficient detail.

REVIEWERS' COMMENTS

Reviewer #1 (Remarks to the Author):

The authors have made significant efforts to address my initial technical concerns. The new results on position dependence of the perturbation are interesting and I'd like to see further development of this in future work (not this article). My only remaining comment is that reference to 'crashing' without the context explained by the authors in their rebuttal still appears in the abstract and should be removed. Otherwise, I think this work is worthy of publication.

Reviewer #2 (Remarks to the Author):

The authors have taken into due consideration all the concerns of the reviewers. As a result, they have rewritten completely the manuscript.

In my first report, I raised a few concerns. After reading this revised version, I am completely satisfied with the changes.

I have no hesitation in recommending publication.

Authors' response to reviews of **Reaching the precision limit with tensor-based wavefront shaping**

R. Gutiérrez-Cuevas, D. Bouchet, J. de Rosny, and S.M. Popoff

27 June 2024

We would like to thank all the reviewers for the time they have spent reviewing our work.

Reviewer 1

Reviewer comment

The authors have made significant efforts to address my initial technical concerns. The new results on position dependence of the perturbation are interesting and I'd like to see further development of this in future work (not this article). My only remaining comment is that reference to 'crashing' without the context explained by the authors in their rebuttal still appears in the abstract and should be removed. Otherwise, I think this work is worthy of publication.

Authors' response

We are glad the reviewer enjoyed our work, and is looking forward for its continuation. In the revised version we have addressed the remaining comment, by removing the reference to "crashing" in abstract.

Modifications

The sentence in question in the abstract has been changed to:

Here, instead, we consider enhancing the interaction between light and perturbations to produce the largest change in the output intensity distribution.

Reviewer 2

Reviewer comment

The authors have taken into due consideration all the concerns of the reviewers. As a result, they have rewritten completely the manuscript.

In my first report, I raised a few concerns. After reading this revised version, I am completely satisfied with the changes.

I have no hesitation in recommending publication.

Authors' response

We are glad the reviewer was satisfied with our replies and now recommends our work for publication.